# Diindoles produced from commensal microbiota metabolites function as endogenous CAR/Nr1i3 ligands

Jiabao Liu [1,10], Ainaz Malekoltojari [1,2,10], Anjana Asokakumar[3], Vimanda Chow[4], Linhao Li [5], Hao Li[6], Marina Grimaldi[7], Nathanlown Dang[3], Jhenielle Campbell[2], Holly Barrett [8], Jianxian Sun [8,9], William Navarre [2], Derek Wilson [4], Hongbing Wang[5], Sridhar Mani [6], Patrick Balaguer[7], Sayeepriyadarshini Anakk [3], Hui Peng [8,9,11] ✉ & Henry M. Krause [1,2,11] ✉

Numerous studies have demonstrated the correlation between human gut bacteria and host physiology, mediated primarily via nuclear receptors (NRs). Despite this body of work, the systematic identification and characterization of microbe-derived ligands that regulate NRs remain a considerable challenge. In this study, we discover a series of diindole molecules produced from commensal bacteria metabolites that act as specific agonists for the orphan constitutive androstane receptor (CAR). Using various biophysical analyses we show that their nanomolar affinities are comparable to those of synthetic CAR agonists, and that they can activate both rodent and human CAR orthologues, which established synthetic agonists cannot. We also find that the diindoles, diindolylmethane (DIM) and diindolylethane (DIE) selectively up-regulate bona fide CAR target genes in primary human hepatocytes and mouse liver without causing significant side effects. These findings provide new insights into the complex interplay between the gut microbiome and host physiology, as well as new tools for disease treatment.

Although many microbe-derived metabolites have been shown to provide major contributions to host metabolism, fitness and health[1–3], most of the molecules responsible, and their underlying mechanisms, are largely unknown. An abundance of studies have revealed that these microbial products can act both locally and systemically by modulating host gene expression and physiological outcomes. One of the known intermediaries in microbial metabolite actions is the constitutive androstane receptor (CAR, Nr1i3), an orphan nuclear receptor

that plays crucial roles in liver metabolism and intestinal inflammation[4–7]. Previous studies have shown that CAR expression and activity in the large intestine is dependent on the presence of gut flora[8]. Conversely, the absence of CAR leads to a decrease in the richness and composition of microbiota[9,10]. A recent study also found that immune cells that infiltrate the small intestinal mucosa rely on CAR to increase expression of the drug transporter MDR1, along with other drug metabolizing enzymes, thereby protecting against bile acid toxicity

[1]Donnelly Centre for Cellular and Biomolecular Research, University of Toronto, Toronto, ON M5S 3E1, Canada. [2]Department of Molecular Genetics, University of Toronto, Toronto, ON, Canada. [3]Department of Molecular and Integrative Physiology, University of Illinois Urbana-Champaign, Urbana, IL 61801, USA. [4]Department of Chemistry, York University, Toronto, ON M3J 1P3, Canada. [5]Department of Pharmaceutical Sciences, University of Maryland School of Pharmacy, 20 Penn Street, Baltimore, MD 21201, USA. [6]Department of Molecular Pharmacology; Department of Genetics; Department of Medicine; Albert Einstein College of Medicine, Bronx, NY 10461, USA. [7]Institut de Recherche en Cancérologie de Montpellier (IRCM), Université Montpellier, Institut régional du Cancer de Montpellier (ICM), Montpellier, Inserm U1194, France. [8]Department of Chemistry, University of Toronto, Toronto, ON M5S 3H6, Canada. [9]School of the Environment, University of Toronto, Toronto, ON M5S 3H6, Canada. [10]These authors contributed equally: Jiabao Liu, Ainaz Malekoltojari. [11]These authors jointly supervised this work: Hui Peng, Henry M. Krause. ✉e-mail: hui.peng@utoronto.ca; h.krause@utoronto.ca

and intestinal inflammation[11]. However, the mechanisms by which these signals are exchanged are currently unknown. Here we identify and characterize a set of indole dimers, referred to as diindoles, as potent CAR agonists. Diindoles are produced by bridging two microbe-derived indole groups with various aldehydes. We go on to show that two of these diindoles, diindolylmethane (DIM) and diindolylethane (DIE), upregulate CAR target gene expression in human primary hepatocytes and mouse liver. Unlike the strong mouse CAR (mCAR) agonist 1,4-bis-[2-(3,5,-dichloropyridyloxy)] benzene (TCPOBOP), side effects such as liver proliferation and enlargement[12] were not observed with DIM or DIE treatment. These differences in toxicity are likely due to differences in the ways that the synthetic agonists and diindoles bind and modulate CAR activity.

## Results

### Affinity pull-downs identify diindoles as CAR ligands

To facilitate affinity co-purification of ligand-bound CAR receptor, we generated a recombinant His-tagged hCAR ligand binding domain (LBD) polypeptide. As previously shown with the related receptor pregnane X receptor (PXR)[13], addition of the LxxLL interaction motif of steroid receptor coactivator-1 (SRC-1) to the hCAR LBD C-terminus rendered the fusion protein stable and soluble when expressed in recombinant bacteria. To validate the function of the purified fusion protein, we tested it using fluorescence thermal shift assays (TSA) with two known agonists (6-(4-chlorophenyl)imidazo[2,1-b][1,3]thiazole-5-carbaldehyde-O-(3,4-dichlorobenzyl)oxime (CITCO)[14] and bisphenol A (BPA)[15], an antagonist (PK11195)[16] and the PXR-specific agonist SR12813. Stabilization by the known CAR ligands only (Supplementary Fig. 1) showed that the fusion protein is highly selective and functional.

Next, the fusion protein was used to test for potential interactions with human microbial metabolites using an extract derived from a complex mixture of microbes cultured from human stool samples[17]. A small amount of the media extract was spiked into an *E. coli* lysate containing 1 μM of the His-tagged CAR LBD-SRC1 fusion protein. After co-incubation, bound metabolites were extracted from the affinity-purified protein and analyzed by untargeted liquid chromatography-mass spectrometry (LC-MS). As PXR is the closest NR to CAR in terms of LBD structure and bound ligands, with significant overlap in ligand-binding profiles, parallel pull-downs were conducted using a PXR LBD-SRC1 fusion protein. This allowed us to identify CAR-specific ligands with little or no affinity for PXR.

Five LC-MS features identified under electrospray ionization positive (ESI$^+$) mode LC-MS were consistently and selectively enriched by the tagged CAR-LBD (mass/charge ratios ($m/z$) = 130.0652, 144.0808, 158.0964, 172.1120 and 186.1277; Fig. 1a, c). Addition of only the DMSO vehicle to the *E.coli* extract showed that the interacting molecules are not present in the unspiked *E.coli* lysate (Fig. 1b). The serial increases in identified hit masses by ~14 daltons each suggest successive additions of $CH_2$ groups to the smallest 130.0652 mass. Alternative potential identities of the $m/z$ 130.0652 metabolite are 3-methylene-indoline or quinoline. Since 3-methylene-indoline is an unstable metabolic intermediate of tryptophan, and a purchased standard of quinoline did not match the metabolite elution time using reverse phase chromatography (Supplementary Fig. 2a), they were dismissed as likely candidates. We next hypothesized that the observed mass feature could have been generated by $[M + H-H_2O]^+$ adduct formation, such as might be produced with indole-3-carbinol (I3C, $C_9H_9NO$). However, the retention time of a purchased I3C standard also failed to match that of the metabolite sample (Supplementary Fig. 2a). Upon further consideration of the small size of these mass features (130–187 amu) as compared to other CAR ligands (250–500 amu)[18], we considered the possibility that the observed masses were in-source fragment ions generated from larger parental molecules that experienced fragmentation during primary ionization.

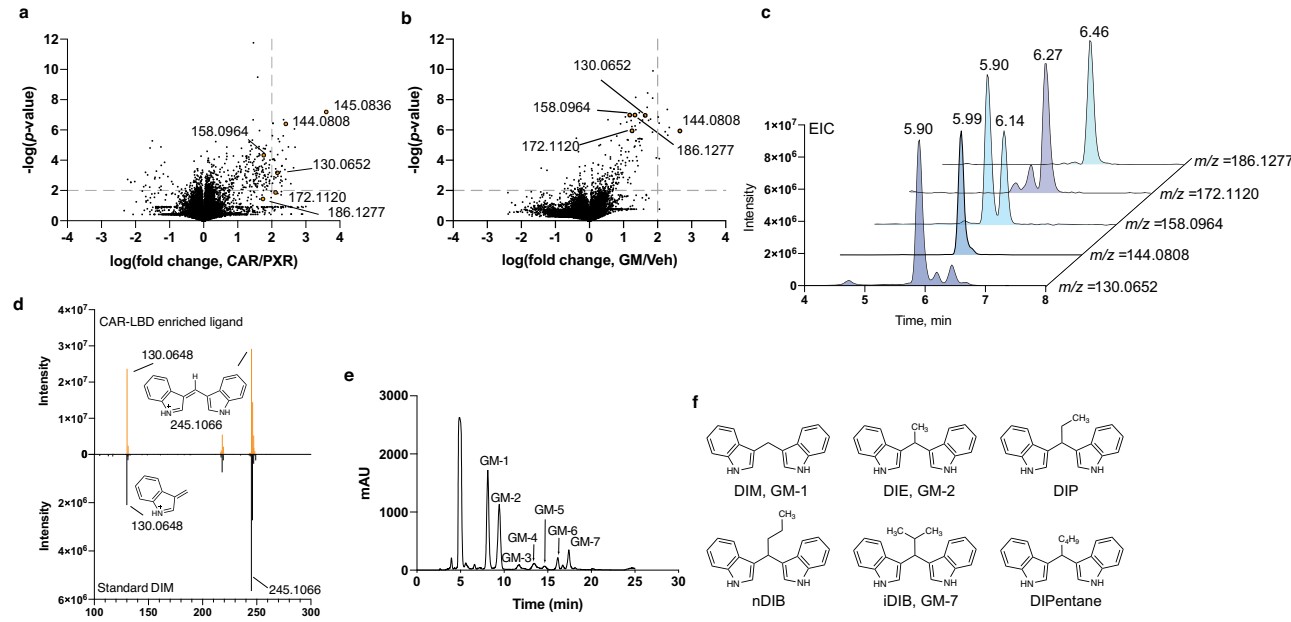

**Fig. 1 | Identification of diindoles as hCAR ligands. a** Volcano plot of the distribution of mass features differentially enriched by CAR in comparison to PXR by affinity purification. **b** Volcano plot of the distribution of mass features differentially enriched by CAR from gut microbiome sample in comparison to vehicle (DMSO). *p* values were determined by two-tailed unpaired Student's *t* test. Dashed lines represent significance cut-offs in analysis (*p* value ≤ 0.01, fold change ≥100). **c** Extracted chromatograms of select CAR-enriched mass features that are separated in mass by increasing numbers of −CH$_2$ groups. **d** MS/MS spectra of CAR-enriched ligand from gut microbiome sample (top panel, orange) compared to that of a purchased DIM reference standard (bottom panel, black). Structures of the fragments $C_{17}H_{13}N_2$ (thereotical $m/z$ 245.1073) and $C_9H_8N$ (thereotical $m/z$ 130.0651) are shown. **e** UV chromatogram of the diindoles-containing subfraction. **f** Structures of diindoles identified by hCAR pulldowns. GM gut microbiome, DIM diindolylmethane, DIE diindolylethane, DIP diindolylpropane, nDIB diindolyl-n-butane, iDIB diindolyl-iso-butane, DIPentane diindolylpentane.

Although the low voltage used in ESI (3–4 kV) to convert neutral molecules into charged ions is meant to preserve the intact structure of the molecule, we hypothesized that in-source fragmentation of the metabolites might nevertheless produce an undetected neutral fragment and [M + H] adducts (diagrammed in Supplementary Fig. 2b). Indeed, the diindole compound DIM has been shown to produce a neutral indole fragment and an [M + H] ion with $m/z$ 130.0 when analyzed by tandem MS/MS[19]. The identity of DIM as the parent source was confirmed by comparing retention times (Supplementary Fig. 2a), in-source fragmentation (Supplementary Fig. 2c), and MS/MS spectra (Fig. 1d) to a purchased DIM standard. As with the CAR pull-down metabolite, the purchased standard produced a low-intensity [M + H] adduct with $m/z$ 247.1223 (Supplementary Fig. 2c), and MS/MS fragmentation of $m/z$ 247.1223 produced the same 130.0652 and 245.1066 product ions (Fig. 1d). This result suggests that the most plausible identities of the other successively larger hits are most likely due to successive additions of $CH_2$ groups to the central methylene, designated as diindolyl-ethane (DIE), -propane (DIP), -butane (DIB), and -pentane (DIPentane) (Supplementary Fig. 2d).

To facilitate structural validation and bioactivity assessments for these previously uncharacterized compounds, we performed a scaled-up isolation of the metabolites from crude human gut microbiome (GM) samples based on our previously observed retention times and masses. Crude GM extract (1 gram) was subjected to normal- and then reverse-phase chromatography yielding 7 pure compounds, termed GM-1–7 (Fig. 1e). GM-1, -2, and -7 were sufficiently abundant for identification by nuclear magnetic resonance (NMR). Using two-dimensional NMR (Supplementary Table 1), GM-1 was confirmed as DIM, GM-2 as 3,3'-diindolylethane (DIE), and GM-7 as 3,3'-diindolyl-isobutane (iDIB) (Fig. 1f).

We also performed customized syntheses of nDIB and the predicted 158.0964 mass ion, 3,3'-diindolylpropane (DIP). Comparing the retention times of the affinity-purified and custom synthesized molecules confirmed the identities of these diindoles, with the exception of the alternative isoforms of iDIB and nDIB, which cannot be distinguished from one another by LC-MS (Supplementary Fig. 3). However, these purifications provided sufficient levels of each molecule for the confirmatory binding and activity assays that follow. The exception was the largest $m/z$ 186.1277 ion, which is most likely diindolylpentane (DIPentane). However, the extreme hydrophobicity and four potential isomers of DIPentane discouraged us from pursuing it further.

## Non-enzymatic diindole production

DIM has been shown to be produced by two pathways in vivo – the I3C pathway and an indole-aldehyde pathway (Fig. 2a)[20]. The I3C pathway is unlikely for the other diindoles, however, as indoyl alcohols cannot form the necessary electrophilic alkylene-indolenine intermediates (Supplementary Fig. 2a) required to nucleophilically attack the second indole. While previous studies have not yet identified diindoles other than DIM in the human gut, DIE and DIB have been found in the marine bacteria *Vibrio parahaemolyticus*[21] and *Pseudovibrio denitrificans*[22], suggesting that microbe production is still a possibility under certain conditions and perhaps detectable with more focused assays. An alternative possibility is that the identified diindoles are generated outside of bacteria via sequential Friedel-Crafts reactions on diet or microbe-derived precursors (Fig. 2a). To test this possibility, indole was combined at 37 °C at multiple pHs with the aldehydes acetaldehyde, propanal, n-butanal or t-butanal. Subsequent LC-MS analysis confirmed that DIE, DIP, and both DIBs were efficiently generated in a pH-dependent manner from indole and corresponding aldehydes (Fig. 2b). Diindole synthesis was most efficient at pH 4–6, with waning production at higher pHs (Fig. 2b). In terms of the sources of indole and aldehydes in the gut, several bacterial species have been shown to convert tryptophan into indole[23], and endogenous aldehydes can also be generated by bacteria in the gastrointestinal tract[24,25], consistent with the potential for efficient diindole production within the gut environment.

## Characterization of CAR-diindole interactions

To ascertain the nature of the physical interactions between diindoles and the CAR LBD, we first employed fluorescence TSAs. In the absence of ligand, a sigmoidal melting curve of the CAR LBD-SRC1 polypeptide was observed, with a melting temperature inflection point ($T_m$) of approximately 51.5 °C (Fig. 3a). When incubated with 0.4–50 μM of each diindole, the protein exhibited dose-dependent shifts in $T_m$ (Supplementary Fig. 4). The thermal shift profiles obtained were similar to those produced using the published CAR ligands BPA and PK11195 (Supplementary Fig. 1), ranging from 6.2 to 8.0 °C, with DIE having the strongest effect (Fig. 3a). Notably, as has been observed previously[26], the synthetic CAR ligand CITCO had no effect on thermal stability (Supplementary Fig. 1), indicating a different non-stabilizing binding mode.

We then used isothermal calorimetry (ITC) to confirm and further characterize the thermodynamics of CAR diindole interactions, and to explore their structure–activity relationships. iDIB was not tested due to solubility issues in this assay. Titrations of solutions of the remaining diindoles, using the His-tagged CAR-LBD-SRC1 fusion protein, resulted in exothermic binding events (Fig. 3b–e, upper panel). When total heat changes were plotted against increasing concentrations of protein, a binding stoichiometry of approximately 1:2 (mol/mol) protein:ligand

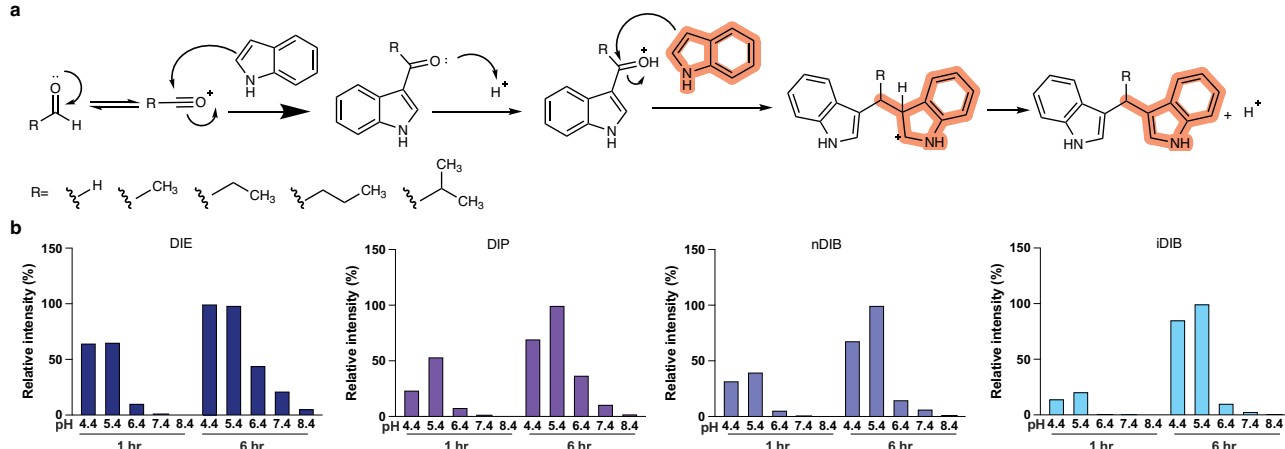

**Fig. 2 | Non-enzymatic production of diindoles. a** Proposed indole-aldehyde pathway underlying the incorporation of aldehydes (R is the side chain for formaldehyde, acetaldehyde, propanal, n-butanal, and i-butanal) into DIM, DIE, DIP, nDIB, and iDIB. **b** pH- and time-dependent diindole production. Values are given as relative intensity,%.

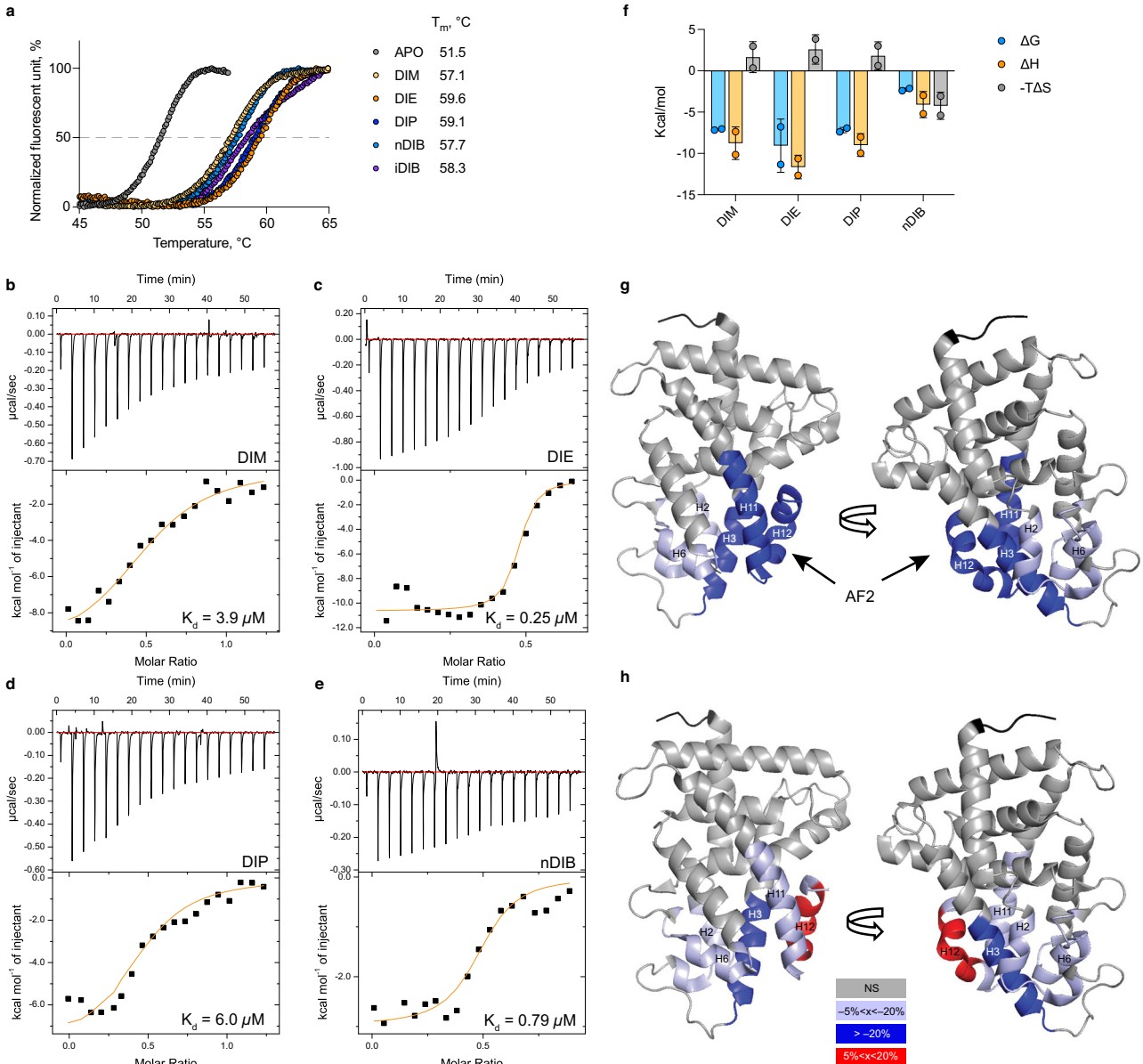

**Fig. 3 | Analysis of diindole interactions with hCAR. a** Thermal shifts induced by diindoles at 12.5 μM (values are means based on triplicates). **b−e** Isothermal titration calorimetry (ITC) characterization of the CAR-SRC1 interaction with DIM (**b**), DIE (**c**), DIP (**d**), or nDIB (**e**). **b−e** Representative heat thermograms (upper panel) and corresponding binding isotherms (low panel). $K_d$ values expressed as the mean of two independent experiments. **f** Respective thermodynamic signatures of binding to the CAR-LBD-SRC1 ($n = 2$ replicates) and represent the mean (±s.d.). **g, h** Differential hydrogen-deuterium exchange analysis of hCAR revealing binding interactions for CITCO and DIE. Structural illustrations mapping the differential HDX data for hCAR upon binding to (**g**) CITCO and (**h**) DIE (PDB code 1XVP) are shown. Structures are color coded according to the summed relative fractional uptake differences in percentage of deuterium incorporation (%D). Dark blue denotes differences >20%, light blue denotes weaker conformational changes (5%< x < 20%) and red denotes increased deuterium uptake. APO apo-CAR-LBD, AF2 activation factor-2.

was inferred for all four compounds (Fig. 3b−e, low panel). Curve fitting yielded dissociation constants ($K_d$s) of 3.9, 0.25, 6.0, and 0.78 μM for DIM, DIE, DIP and nDIB, respectively (Fig. 3b−e). The Gibbs free energy of reaction was favorable for each compound, particularly so for DIE (Fig. 3f), indicating that ligand-binding is spontaneous. Diindole ligand binding was also characterized by a beneficial negative binding enthalpy ($\Delta H$). However, the favorable enthalpy values for DIM, DIE, and DIP were also accompanied by a significant loss of entropy ($\Delta S$), indicative of a loss of conformational freedom associated with ligand binding[27]. The thermodynamic responses for nDIB indicate both favorable enthalpy and entropy contributions, which together provide a comparable Gibbs energy ($\Delta G$) to DIE (Fig. 3f) consistent with their similarly high affinities ($K_d$s = 0.78 and 0.25 μM, respectively; Fig. 3c, e).

To gain further insight into the spatial diindole binding contacts and effects on the CAR LBD structure, we also employed hydrogen-deuterium exchange coupled with MS (HDX-MS) analysis. HDX-MS has previously been used to determine the conformational dynamics of mCAR LBD regions induced upon binding of the mCAR synthetic ligand TCPOBOP[28]. Here, we used the highest affinity diindole ligand DIE to compare the rate and location of deuterium exchange within the hCAR LBD compared to the effects of CITCO, the high-affinity hCAR synthetic ligand. All HDX-MS experiments were conducted for 1, 10, and 30 min and for three states: hCAR in the absence of any ligand (apo hCAR), in the presence of 100 μM CITCO, and in the presence of 100 μM DIE (Supplementary Table 2). The average sequence coverage for all experiments was 94.1% with an average peptide redundancy of

5.49. For both CITCO and DIE, amino acids 51–65 (H3) displayed the most substantial difference in deuterium uptake (>20%) compared to apo hCAR. Both ligands also induced a relatively small (10%–20%) decrease in deuterium uptake for amino acids 28–39 (H2) and 115–133 (H6), indicating a relatively subtle change in structure or dynamics upon ligand binding (Fig. 3g, h).

These results show that DIE binds to hCAR with many of the same structurally determined interaction sites as CITCO (at the interface of helices 2, 3, and 6). However, two regions of the CAR LBD responded differently to the two ligands. For amino acids 217–234 (H11), complexation with CITCO resulted in a more pronounced deuterium uptake (>20%, Fig. 3g) compared to complexation with DIE (<10%, Fig. 3h). In contrast, for adjacent residues 235–243 (H12/AF2) CITCO binding decreased deuterium uptake while DIE increased uptake. This indicates that, despite the general position of helix 12 in a constitutively active conformation, CITCO is still able to further stabilize this conformation. This stabilization of the cofactor docking site is consistent with effects on cofactor recruitment. A time-resolved fluorescence resonance energy transfer (TF-FRET)-based proliferator-activated receptor gamma coactivator 1-alpha (PGC1α) cofactor recruitment assay shows that CITCO is able to increase PGC1α recruitment beyond 'constitutive' levels, whereas DIE does not (Supplementary Fig. 5).

### Diindoles selectively activate hCAR transcriptional activity

As noted above, the non-canonical fixed structure of hCAR LBD helices 10–12 renders it constitutively active when in the nucleus. However, unliganded CAR is normally kept inactive by retention in the cytoplasm[29], with ligand binding promoting nuclear transport. To evaluate the ability of diindoles to promote hCAR nuclear translocation, we conducted an intracellular translocation assay using primary human hepatocytes (PHHs). Using PHHs obtained from two individuals, DIM and DIE were shown to efficiently promote hCAR nuclear relocation (Fig. 4a).

Although CAR activity is considered to be constitutive once in the nucleus, we used a couple of other transcriptional approaches to test for nuanced diindole effects. The first makes use of an antagonist displacement-based assay. A GAL4-hCAR LBD construct was transfected into HEK293 cells together with the hCAR inverse agonist PK11195 ($IC_{50} = 1.3 \mu M$). Inclusion of the antagonist resulted in an almost 6-fold decrease in luciferase reporter expression (Supplementary Fig. 5a). Co-treatment with PK11195 and $3 \mu M$ CITCO results in an approximately 50% recovery of the reporter signal (Supplementary Fig. 6a). Co-treatment with PK11195 and DIM, DIE or nDIB also resulted in derepression, with $EC_{50}s$ in the micromolar to submicromolar range (Fig. 4b). The $EC_{50}s$ of DIP and iDIB could not be determined in this assay due to elevated non-specific luminescence (Supplementary Fig. 6b). Notably, all diindoles tested exhibited greater efficacy than CITCO, with an $E_{max}$ value -1.5-fold higher than CITCO (Fig. 4b). Given that CITCO has a unique ability to further stimulate coactivator binding, this -1.5-fold advantage in diindole activity suggests a greater ability to displace the inverse agonist.

Our second transcriptional approach made use of an hCAR variant that has an alanine insertion between LBD helices 7 and 8 (hCAR1+A) that disrupts its constitutive activity and makes it responsive to ligands[30]. When transfected together with a CYP2B6 reporter, DIM and DIE were both able to significantly boost hCAR1+A activity (Fig. 4c), suggesting that both can induce activity-promoting conformational changes from within the LBD pocket.

Indole derivatives, including the diindole DIM, have previously been reported to also act as PXR and aryl hydrocarbon receptor (AhR) ligands[31,32]. To test whether the diindoles identified here also bind to PXR or AhR, we tested for their potential effects on PXR and AhR reporters in cultured cells. While some of the diindoles were able to activate PXR (Supplementary Fig. 6c, d), the required concentrations (5–15 μM) far exceed the lower physiological range that activates CAR. No AhR responses were observed at the relatively high tested concentration of 10 μM (Supplementary Fig. 6e).

We then investigated the ability of DIM and DIE to activate transcription of a known direct hCAR target gene, cyp2b6, in PHHs. Consistent with our earlier reporter gene assays, DIM and DIE selectively upregulated cyp2b6 mRNA expression in both donors, while having no effects on the non-target gene sult2a1[33] (Fig. 4d). Effects were also seen on the joint CAR/PXR target gene cyp3a4[34], though these effects tended to be more variable (Fig. 4d). Taken together, our cell-based assays show that diindoles, DIE in particular, act as strong hCAR agonists with almost the same efficacy as the potent synthetic agonist CITCO.

### Diindoles activate mCAR transcriptional activity in vivo

As previously noted, one of CAR's functions is the detoxification of xenobiotics, which is mediated in part by activation of cytochrome P450 (cyp), glutathione S-transferase and sulfotransferase (sult) genes. We looked to see how these genes are affected in mouse livers after a 3-day regimen of DIM or DIE intraperitoneal injections. We found that the gold standard mCAR target gene cyp2b10 (human cyp2b6)[35] is robustly induced in wild-type (WT) mice, but not in CAR-deficient mice, by both DIM and DIE (Fig. 5a and Supplementary Fig. 7a). As a control, we also pre-treated mice with the CAR inverse agonist androstenol (And) to selectively reduce CAR target gene expression[36]. As expected, pre-treatment with And greatly reduced target gene expression and responses to both DIM and DIE (Fig. 5a and Supplementary Fig.7a).

Other previously reported mCAR target genes responded less specifically. For example, cyp3a11 (orthologous to human cyp3a4), which is also a known mPXR target gene[37], showed an increase rather than a decrease in response to And pre-treatment (Fig. 5b), most likely because And is also a mPXR agonist[38]. The subsequent addition of DIM or DIE did not affect this And-dependent induction, nor did their addition in the absence of And (Fig. 5b), suggesting that PXR may be the major regulator of this gene under these conditions. Expression of sult2a1, a previously reported CAR, PXR, AhR, and VDR target gene[33,39,40], also did not show significant responses to DIM, DIE or And in either WT or $Nr1i3^{-/-}$ mice (Fig. 5c, Supplementary Fig.6b, c). These results indicate that DIM and DIE can both selectively promote CAR target gene activation in vivo, though effects are likely complicated by indirect effects, compensatory effects and sex-specific variations.

Mouse treatments with the mCAR synthetic agonist TCPOBOP tend to highly activate mCAR target genes (Supplementary Fig. 7d–f), resulting in robust liver cell proliferation, growth (Supplementary Fig. 8a), toxicity and occasional carcinogenesis[41,42]. To test whether our diindole compounds cause similar outcomes, we first examined post-treatment liver weights. Unlike the growth increases observed after TCPOBOP treatment, no significant changes were observed after DIM or DIE treatment (Supplementary Fig. 8b, c). We then performed histological analyses of liver tissues. Hematoxylin and eosin (H&E) staining revealed no increase in hepatocyte size, cytoplasmic swelling, obvious damage or inflammation after DIM or DIE treatment (Fig. 5d), suggesting that these ligands are well tolerated in the liver. We did observe occasional focal inflammation and mild cholangitis with And treatment, but this was reduced by co-treatment with DIM or DIE (Fig. 5d).

## Discussion

Our identification of a series of diindole compounds as high-affinity CAR ligands will aid in understanding the relatively understudied roles played by CAR in host-gut microbe interactions as well as causes and solutions for associated metabolic, physiological and inflammatory responses. In this study, we focused primarily on the two simplest and most soluble diindoles, DIM and DIE. While DIM has previously been found to have preventative roles in cancer[43], until now, its actions were

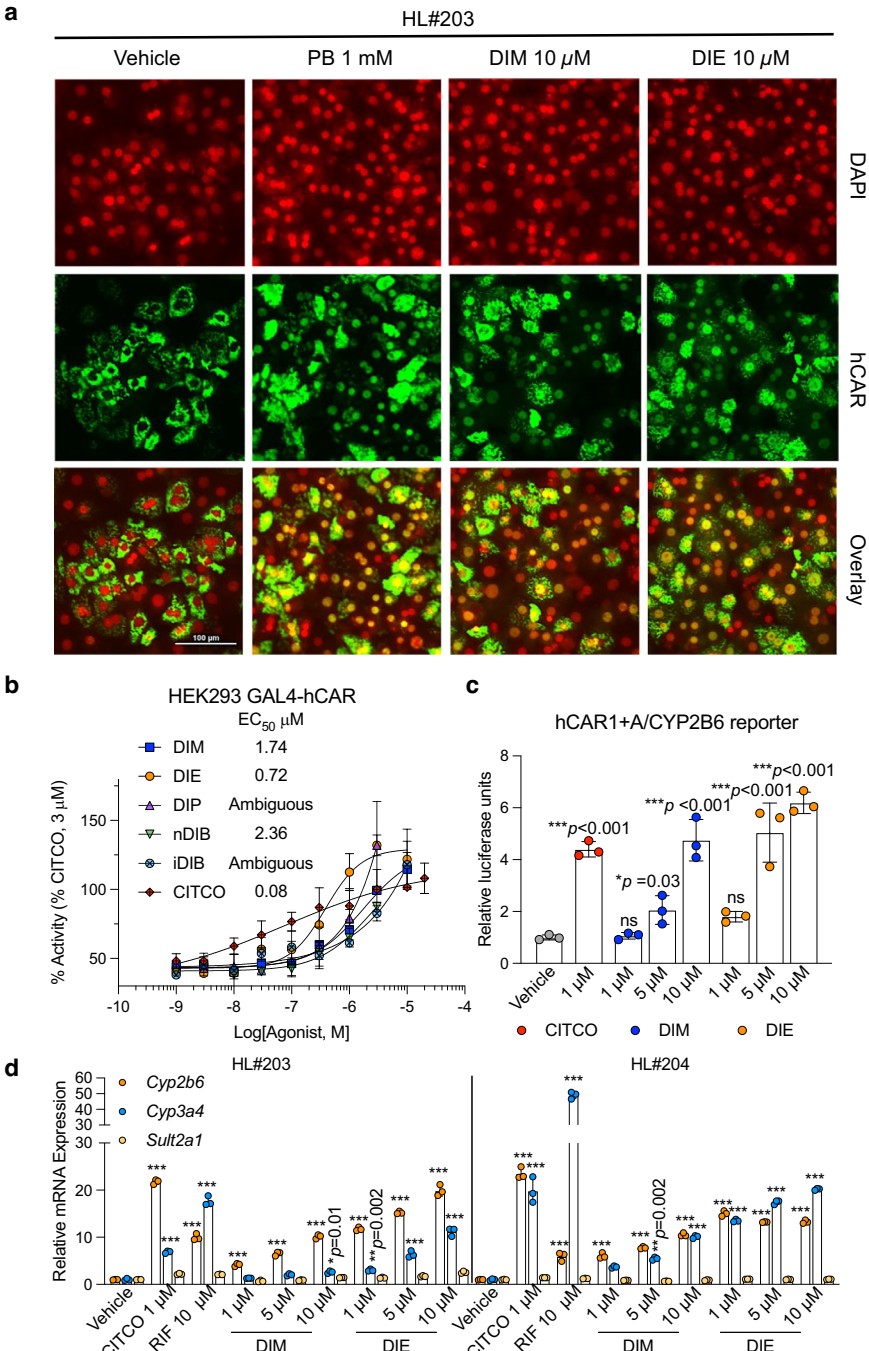

**Fig. 4 | Diindoles induce hCAR nuclear translocation and transcriptional activity. a** Primary human hepatocytes (PHHs) from liver donor #203 were infected with 2 μL of adenoviral enhanced yellow fluorescent protein-tagged hCAR (Ad/EYFP-hCAR) followed by the treatment with 0.1% DMSO, 1 mM PB, 10 μM DIM or DIE for 8 h. **b** GAL4-hCAR-LBD-transfected HEK293 cells were exposed to different concentrations of CITCO, DIM, DIE, DIP, nDIB, or iDIB. Basal reporter activity is reduced with 1 μM PK11195 treatment. Assays were performed in triplicate ($n = 3$ independent experiments), and data are expressed as means (±s.d.). **c** HepG2 cells were transfected with a CYP2B6-2.2 kb reporter and hCAR1+A expression vectors. Transfected cells were subsequently treated with vehicle control (0.1% DMSO), CITCO, DIM, or DIE at indicated concentrations for 24 h. Results represent 2 independent experiments (one-way ANOVA test in DIM and DIE; and two-tailed unpaired Student's $t$-test in CITCO, $n = 3$/each, data expressed as means ± (s.d.). *$p < 0.05$, ***$p < 0.001$ compared to the vehicle). **d** RT-qPCR analysis of *cyp2b6*, *cyp3a4*, and *sult2a1* mRNA expression in PHH (from two donors HL#203 and HL#204) treated for 48 h with vehicle (0.1% DMSO) or indicated concentrations of ligands. Results were obtained from experiments performed in triplicate. Data are expressed as means ± (s.d.); one-way ANOVA test, ***$p < 0.001$ compared to the vehicle. PB phenobarbital.

believed to be mediated by PXR and AhR. However, we show here that DIM is a high-affinity CAR ligand that is unlikely to bind these alternative receptors at physiological concentrations. DIE and the other identified diindoles are also likely to have similar, CAR-dependant disease applications.

In terms of their sources, only DIM has been shown to be microbiome-produced[20]. Our results suggest that the others are most likely or often to be produced from the spontaneous interaction of microbe-produced indoles and aldehydes, which are abundant in the appropriately acidic upper intestine[23,24,44]. Notably, this is where CAR

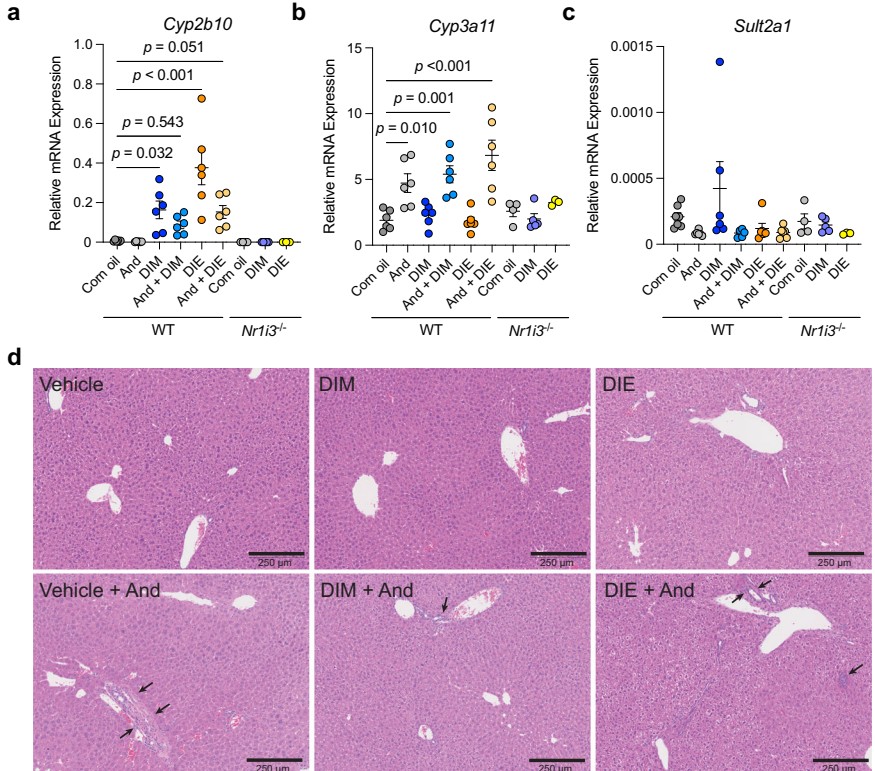

**Fig. 5 | Diindoles activate mCAR target genes.** WT and $Nr1i3^{-/-}$ male mice were treated with Vehicle, DIM, DIE CAR inverse agonist, androstenol as indicated. Graphs show qPCR analyses of genes **a** *cyp2b10*, **b** *cyp3a11*, and **c** *sult2a1*. ($n = 6$ for WT mice, $n = 4$ for corn oil, $n = 5$ for DIM, and $n = 3$ for DIE in $Nr1i3^{-/-}$ mice). Data shown are means ± (SEM) (statistically significant pairwise comparisons were calculated using ordinary one-way ANOVA tests; $p$ values determined comparing to corn oil or And). **d** Representative liver histology images of the treatment groups. Areas of focal inflammation or mild cholangitis are marked with black arrows. And androstenol.

has recently been shown to be required for the prevention of bile acid-induced inflammation[11]. Supplementary Fig. 9 shows that, on their own, lactobacilli, which contribute to the low pH of this region, produce acetaldehyde and, when supplemented with indole, produce high levels of DIE. Indole has already been shown to be produced by several commensal bacterial strains[45].

In addition to microbe-produced acetaldehyde, which is required for DIE production, acetaldehyde can also be produced in the liver from alcohol by the enzyme aldehyde dehydrogenase-2[46]. Thus, DIE also has the potential to be produced in the liver if gut microbe-produced indole is transported there. While the presence of DIE in the liver may have beneficial effects, chronic production as a consequence of prolonged alcohol intake may be harmful[47], given that mCAR activation by prolonged phenobarbital consumption is associated with hepatocellular carcinomas[48].

Our results show that the agonistic effects of diindoles on CAR are different than those mediated by the human synthetic CAR ligand CITCO and the mouse synthetic CAR ligand TCPOBOP. CAR is an unusual NR in that its ability to recruit coactivators is essentially constitutive due to the fixed position of H12. Instead of the classic NR ligand activation method, which involves changes in the LBD pocket structure and H12 repositioning to form a co-activator-binding pocket, CAR ligands act by disrupting chaperoned dimeric complexes in the cytoplasm, allowing translocation of CAR monomers to the nucleus[49]. CITCO and TCPOBOP were not selected based on the latter, but instead on their abilities to further enhance cofactor recruitment[14,50]. Our data, including HDX-MS analyses, suggest that diindoles occupy the hCAR LBD pocket, whereas CITCO binding may primarily occupy a more peripheral site near or affecting the AF-2 pocket. This likely helps explain why CITCO has no effects on hCAR thermal stability in both

TSA (Supplementary Fig. 1c, e)[26] and ITC-based assays. These results are consistent with the abilities of diindoles to bind and activate both hCAR and mCAR, whereas CITCO only binds hCAR and TCPOBOP only binds mCAR.

In both human primary liver cells and mouse livers, DIM and DIE acted as potent activators of the well-established CAR mouse and human orthologue target genes *cyp2b6* and *cyp2b10*. The effects of DIM and DIE on other previously reported CAR target genes in three-day treated mouse livers were less clear. We suspect that some of these are either not direct target genes, or are affected by compensatory effects over the course of treatment. It is worth noting that DIM and DIE are rapidly broken down by CAR target gene induction, as their levels accumulated significantly in $Nr1i3^{-/-}$ mice (Supplemental Fig. 10). These results may also explain why diindole treatments did not result in liver hyperplasia, as is typically caused by TCPOBOP[12]. With higher stability, and potentially broader and higher target gene activation, the actions of TCPOBOP may be more prolonged and diverse.

As noted earlier, CAR plays roles in a number of processes and diseases, including diabetes, fatty liver disease, and various inflammatory disorders[5]. Natural high-affinity ligands such as DIM and DIE may be suitable for treating these disorders. They also may serve as templates for the development of similar compounds with optimized pharmacokinetc properties and targeted outcomes.

## Methods
### Reagents
DIM (15927), SR12813 (18115), I3C (11325), and androstenol (33457) were purchased from Cayman Chemical. CITCO (C6240), TCPOBOP (T1443), PK11195 (C0424), BPA (239658), quinoline (94517), indole (I3408), formaldehyde (252549), acetaldehyde (402788),

propionaldehyde (538124), n-butyraldehyde (418102), and i-butyraldehyde (240788) were purchased from Sigma. DIE (2021-0486967), DIP (2021-0488974), nDIB (2021-0485695), and iDIB (2021-0488971) were purchased from Enamine.

## Animal experiments

12–15 week-old C57BL/6 J mice were obtained from Jackson Laboratories and were housed on a standard 12 h light/dark cycle. $Nr1i3^{-/-}$ mice with C57BL/6 J background were obtained from David Moore's lab (PMID: 11057673) at Baylor College of Medicine, and used at 13–26 weeks of age. Wild-type male, female and $Nr1i3^{-/-}$ male mice were used in this study, due to known sex-specific CAR activities. Mice were housed in groups of 2–5 mice per cage on a standard 12 h light/dark cycle and fed with rodent chow diet (Inotiv, Teklad rodent diet 2918). Temperature was maintained at 21 °C with a range of 21–24 °C and humidity at 16–51%. National Institute of Health guidelines for the care and use of laboratory animals were followed with all experiments performed as directed by the Institutional Animal Care and Use Committee at the University of Illinois, Urbana-Champaign (UIUC, IACUC protocol: 22183). Mice were acclimatized to the UIUC mouse facility for 2 weeks prior to experiments. DIM and DIE were dissolved in 5% DMSO in corn oil and injected intraperitoneally at 25 mg/kg once daily for three days. After 4 h of the third injection, mice were sacrificed to collect serum, livers and intestines. TCPOBOP was dissolved in corn oil and injected intraperitoneally at 3 mg/kg once and the livers collected after 3 days for further studies. When used to block CAR activity, 100 mg/kg androstenol was injected intraperitoneally 15 min prior to the first DIM or DIE treatment[51]. RNA from mice livers was isolated using TRIzol (Invitrogen) reagent using the manufacturer's instructions. 3–5 µg of RNA was used to synthesize cDNA with random primers (NEBioLabs) and Maxima Reverse Transcriptase kits (Thermo Fisher Scientific). cDNA was further diluted, and 50 ng was used for SYBR green (Thermo Fisher Scientific) based qRT-PCR. Relative gene expression was calculated by the delta Ct method and normalized to *36b4* levels as loading controls. qPCR primers for gene expression analysis are listed in Supplementary Table 3.

## Construction of expression plasmids

Human CAR LBD and SRC1 peptides were cloned into a pET28-MHL (GenBank accession EF456735) vector, which was first linearized by BseRI (NEB, R0581S) digestion. The human CAR LBD ($Lys^{108}$ – $Ser^{352}$) was amplified by PCR from cDNA with an N-terminal 15-bp extension complementary to the linearized backbone, and a C-terminal 15-bp extension complementary to the SRC1 fragment. The SRC1 peptide ($Ser^{682}$ – $Ser^{700}$) with a 10-residue upstream linker (N-SGGSGGSSHS-C) was amplified by PCR from a pLIC-PXR_link_SRC1 plasmid generously provided by the Redinbo Laboratory (UNC Chapel Hill). The SRC1 fragment was amplified with an N-terminal 15-bp extension complementary to the hCAR LBD and a C-terminal 15-bp extension complementary to the linearized vector. The DNA fragments were combined and inserted into a pET28-MHL vector by In-Fusion HD Cloning (Takara Bio, CA, USA). The final construct contains a HIS6-tagged hCAR LBD fused to the SRC1 peptide by a flexible 10-residue linker.

A pET28-MHL hPXR LBD-SRC1 expression plasmid was made by amplifying the hPXR LBD-SRC1 cassette from the pLIC vector with 15-bp extensions complementary to each end of the BsERI digested pET28-MHL backbone. The amplified DNA was inserted into the pET28-MHL vector by In-Fusion HD Cloning.

## Pull-downs

hCAR LBD-SRC1 and hPXR LBD-SRC1 vectors were transformed into BL21(DE3) cells (NEB, C2527) and cultures grown in LB medium at 37 °C until an $OD_{600}$ of 1.0 was reached. The temperature was then decreased to 17 °C, and cultures were induced with 0.1 mM isopropyl 1-thio-$\beta$-D-glucopyranoside (IPTG, BioShop IPT001) for 16 hrs. Harvested bacterial pellets were resuspended in buffer containing 20 mM Tris/HCl (pH 8.2), 300 mM NaCl, 5 mM imidazole (BioShop, IMD508), and 0.5 mM DTT (BioShop, DTT001). After sonication (5 min at 40% amplitude using cycles of 10 s-on/10 s-off) and centrifugation at 18,000 × g at 4 °C for 20 min, supernatants were incubated with gut microbiome samples (1:100, v-v) and 7 µL of HIS-select nickel magnetic agarose beads (H9914, Sigma-Aldrich) for 30 min at 4 °C, then for 15 min at room temperature. Flowthroughs were removed using a magnetic stand. After washing three times with 100 µL of buffer containing 20 mM Tris/HCl (pH 8.2), 300 mM NaCl and 30 mM imidazole, the His-tagged LBD-ligand complexes were eluted with 100 µL of buffer containing 20 mM Tris/HCl (pH 8.2), 300 mM NaCl and 300 mM imidazole using a magnetic stand.

## LC-MS analysis

LC-MS sample preparation and data analysis were performed as described in a previous publication[52], with minor modifications. Elution samples ($n = 3$) were loaded onto a Zeba spin desalting plate (Thermo Fisher Scientific, 7 K MWCO (89807)), then centrifuged at 1500 g at 4 °C for 2 min. The desalted samples were then evaporated by SpeedVac and resuspended in HPLC grade methanol (Fisher Chemical A4564). Samples were then loaded onto a reverse phase column (Accucore™ Vanquish C18+, 1.5 µm 50 × 2.1 mm, Thermo Fisher) using a UPLC system in methanol, and eluted with a gradient starting with 90% mobile phase A ($H_2O$ (Fisher Chemical 022934-M6) containing 0.1% formic acid (Fisher Chemical A11750)) and 10% mobile phase B (methanol containing 0.1% formic acid) at time zero, with a flow rate of 0.15 mL/min, then increasing to 50% mobile phase B at 2 min, 100% mobile phase B at 5 min, then kept isocratic for 3 min before returning to 90% mobile phase A over the next 0.5 min, and finally kept isocratic for 1.5 min. The temperatures of the column and sample compartments were maintained at 40 °C and 10 °C, respectively. Data were acquired in full MS and parallel reaction monitoring (PRM) mode. Spray voltage (+) was 3.5 kv and capillary temperature (+ or +−) was 300 °C. Parameters for full MS scan were recorded at resolution $R = 70,000$ with a maximum of $3 × 10^6$ ions collected within 200 ms, followed by PRM MS/MS scan recorded at resolution $R = 17,500$ (at $m/z$ 247.1229) with a maximum of $3 × 10^6$ ions collected within 200 ms. Untargeted metabolomics data analysis was performed using an in-house R program. Specifically, raw MS files were converted to open format (.mzXML). Chemical features were detected with XCMS package at 2.5 ppm mass accuracy. Features were matched across different samples with a 0.5 min retention time tolerance window. Putative metabolite features were selected by calculating the fold change of features from treatments relative to controls. Only those features with 100-fold higher abundance ($p < 0.01$) using CAR LBD versus controls were considered true metabolite features.

## hCAR recombinant protein production

Recombinant proteins were expressed as described above in the pulldown section. Proteins were purified using of HisPur Ni-NTA resin (Thermo Fisher Scientific, 88222), followed by size exclusion FPLC (HiLoad 16/60 Superdex 200, GE Healthcare, Life Sciences) equilibrated with 150 mM NaCl, 50 mM HEPES (pH 8.2) and 0.5 mM DTT at a flow rate of 1 mL/min.

## LC-MS guided sample fractionation

1 gram of crude GM extract was applied to a column packed with 10 g silica and washed with chloroform ($CHCl_3$, Fisher Chemical, BPC298500). Sample was serially eluted into five fractions using 1 mL mobile phase comprised of $CHCl_3$ and an increasing gradient of MeOH. Each fraction was analyzed by LC/MS to identify fractions enriched for the mass features of interest. A $CHCl_3$:MeOH solution of 100:1 yielded the fraction with greatest enrichment. This fraction was further

purified into eight fractions by reverse phase C8 chromatography (Luna 5 μm C8(2) 100 Å, 250 × 10 mm, Phenomenex) using a gradient elution of 20% $H_2O$ and 80% MeOH, increasing to 100% MeOH over 20 min. Each fraction was analyzed by LC/MS to confirm the presence and relative yield of target compounds. Column temperatures and sample compartments were maintained at 20 °C and 4 °C, respectively. Data were acquired at a wavelength of 280 nm using a DAD detector.

## NMR analysis of isolated GM fractions

$^1$H and $^{13}$C NMR spectra were recorded on a 700 MHz Agilent DD2 NMR Spectrometer (Agilent, Santa Clara CA, USA). Chemical shifts are given in $\delta$ (ppm) values relative to those obtained from the solvent signal [MeOH-$d_4$ ($\delta_H$ 4.78; $\delta_C$ 49.2)]. Standard pulse sequences programmed into the instrument were used for each measurement.

## De novo diindole synthesis

A 1 mL solution of indole (0.25 mM) was combined with formaldehyde, acetaldehyde, propionaldehyde or n-/i-butyraldehyde (0.2 mM) in water. The mixtures were stirred in sealed tubes at 37 ± 2°C at pH 4.4, 5.4, 6.4, 7.4, or 8.4 for 1 and 6 h. Each reaction mixture was subjected to LC-MS analysis to determine the abundance of di-indole products formed.

## Lactobacillus diindole production

All bacterial culturing took place in an anaerobic chamber at 37 °C. Frozen bacterial stock of each strain was streaked on MRS agar plates and incubated for 2 days. Strains were re-streaked on fresh plates and incubated an additional 2 days. 10 mL of MRS broth was inoculated with a single colony of each strain and incubated overnight. 1 μL of each overnight culture was added to wells of a 96-well round bottom tissue culture plate in triplicate containing either 99 μL of MRS or MRS supplemented with 500 μM indole. Plates were covered, sealed and incubated for 48 h. Bacteria were resuspended in media by gently pipetting up and down in each well. $OD_{600}$ readings were taken using an Infinite Pro 200 Microplate Plate Reader (Tecan, Männedorf, Switzerland). Lipids were extracted from the culture using a $CHCl_3$:MeOH:$H_2O$ (4:5:1) mixture and spun down to separate organic and non-organic layers. The lower chloroform layer containing the diindole compounds was carefully transferred to a fresh plate, evaporated under vacuum, then resuspended in 100 μL of methanol. Plates were stored at −20 °C until LC-MS analysis.

## LanthaScreen™ TR-FRET hCAR coactivator assays

LanthaScreen™ TR-FRET hCAR coactivator assays were performed according to manufacturer's instructions (Invitrogen, A15141). Test compounds were incubated for 2 h at room temperature with GST-hCAR-LBD (5 nM), terbium-labeled anti-GST antibody (5 nM), and Fluorescein-PGC1α coactivator peptide (125 nM). Incubations were performed in four technical replicates. TR-FRET emissions were measured using a Tecan Infinite F200 Fluorescence Microplate Reader. Results were expressed as the ratio of fluorescence intensity at 520 nm (fluorescein emission excited by terbium emissions) and 495 nm (terbium emissions).

## Fluorescence TSAs

Purified hCAR LBD-SRC1 protein was assayed in 137 mM NaCl, 2.7 mM KCl, 10 mM $Na_2HPO_4$, and 1.8 mM $KH_2PO_4$, 0.5% DMSO, pH 7.4, at a final concentration of 2.2 μM. SYPRO Orange was used at a final concentration of 10× (Invitrogen, S6650). Each sample was divided into three 25 μL replicates. Sample solutions were dispensed into 96-well optical reaction plates (Thermo Fisher Scientific, 4306737) and sealed with optical PCR plate sheets (Thermo Fisher Scientific, AB-1170). TSA experiments were carried out using a QuantStudio 12 K Flex Real Time PCR System (Applied Biosystems). The temperature was ramped up from 25 °C to 95 °C at a rate of 0.3 °C/min, and SYPRO orange

fluorescence monitored. Raw fluorescence data and melting temperatures were determined using nonlinear regression curve fitting using GraphPad Prism 9.

## Isothermal titration calorimetry

Purified hCAR-SRC1 protein (200–250 μM) was dissolved in 120 μL of PBS and supplemented with 0.5% DMSO. All experiments were performed in a MicroCal Auto-iTC200 instrument (Malvern Panalytical, Marlvern, United Kingdom) at 25 °C. Ligands were used at a concentration of 50 μM. Protein was titrated into cells in a series of 19 injections (injection 1 = 0.4 μL, injection 2–19 = 2 μL) 180 s apart. Control experiments performed by injection of the potein into ligand solution yielded insignificant heats of dilution, and heat responses were subtracted. Integrated heat effects were analyzed using Microcal Origin 7 software (Malvern Instruments).

## Bottom-up HDX-MS

Human CAR protein was prepared at 10 μM in 100 mM ammonium acetate pH 7.25 and diluted into either equilibration buffer (10 mM phosphate buffer, pH 7.5) or deuterated reaction buffer (10 mM phosphate buffer, 150 mM NaCl, pH 7.5 in $D_2O$). After labeling times of 1, 10 or 30 min at 25 °C, samples were mixed with equal volumes of quench buffer (100 mM phosphate buffer, pH 2.5) at 0 °C. For protein-ligand complex samples, hCAR was mixed 1:10 with CITCO or DIE compound at a final concentration of 100 μM with 2% DMSO. HDX-MS was performed using a commercial Waters HDX system (M-class ACQUITY UPLC and Water Cyclic IMS Mass Spectrometer). Leucine enkephalin was used for mass correction, and all time points were carried out in triplicate. Samples were digested using an Enzymate BEH Pepsin column (2.1 × 30 mm, Waters) at 15 °C. Peptides were desalted on an ACQUITY UPLC BEH C18 VanGuard Pre-column (2.1 × 5 mm, Waters) and separated using an ACQUITY UPLC BEH C18 column (100 × 1 mm, Waters). ProteinLynx Global server (PLGS) and DynamX software was used for peptide identification and deuterium uptake analysis, respectively. PyMOL 2.5 was used for structural visualization.

## Mammalian reporter gene plasmids

The GAL4-hCAR, UAS-Luc, pcDNA-GAL4, pGL4-Renilla-Luc and pGEM plasmids were generously provided by Dr. David Mangelsdorf. The GAL4-hCAR plasmid encodes a constitutively expressed chimeric yeast GAL4 DBD fused to an hCAR LBD. The UAS-Luc plasmid encodes a Firefly luciferase reporter under the control of a GAL4 response element. The pcDNA plasmid encodes a constitutively expressed GAL4 DBD and was used as a control vector to determine non-specific luminescence in all HEK293T cellular reporter assays. The pGL4-Renilla-Luc plasmid encodes a constitutively expressed Renilla luciferase reporter and was used as a transfection control in the HEK293T cellular reporter assays. The pGEM plasmid does not contain any mammalian expression cassettes and was used as a filler plasmid for transfections.

## HEK293T dual transfection reporter gene assays

HEK293T cells (ATCC, CRL-3216) were seeded at a density of 40,000 cells per well in white flat-bottom 96-well microplates (Greiner Cell-Star, 655073) in Dulbecco's Modified Eagle Medium (DMEM) (2.5 g/l glucose) supplemented with 10% stripped fetal bovine serum with 100 units/ml of penicillin and incubated for 24 h at 37 °C. Transfection was carried out using 0.3 μL/well of Lipofectamine 2000 (Thermo Fisher Scientific, 11668019). Cells were transfected with 10 ng/well of either pcDNA or GAL4-hCAR, 50 ng/well of UAS-Luc, 5 ng/well of pGL4-Renilla-Luc, and 85 ng/well of pGEM for a total of 150 ng/well of DNA. Following a brief incubation (5 h), media was gently aspirated from each well, and test compounds were added in media supplemented with 1 μM PK11195. Cells were incubated for 16-hours to allow build-up of luciferase and then processed using the Dual-Glo Luciferase Assay

System (Promega, E2940) as per the manufacturer's instructions. Incubations were performed in four technical replicates.

PXR transfection assays were performed in HEK293T cells as previously published with the following modifications[32]. 30,000 cells per well in 96 well-plate were transfected using 200 ng/well of Gal4-hPXR-LBD plasmid, 100 ng/well of β-Gal expression plasmid, and 700 ng/well of TK-(MH100)4-LUC reporter. After 24 h transfection using Lipofectamine 2000 (Thermo Fisher Scientific, 11668019), cells were exposed to testing compounds for 24 h. A Neolite Reporter Gene Assay System (PerkinElmer, 6016716) was used to determine luciferase activities using a plate reader (Perkin Elmer, X5 2030 Multilabel Reader). Data were expressed as mean ± SD RLU (relative light units from luciferase assay) normalized to β-Gal activity in the same well.

### HG5LN stable reporter gene assays
HG5LN, HG5LN hCAR, and HG5LN hPXR cell lines were established as previously described[53,54]. Briefly, the HG5LN cell line was established by integration of a GAL4-responsive luciferase reporter gene (GAL4RE5-bGlob-Luc-SV-Neo) in HeLa cells[53]. HG5LN cells were then stably transfected using a pSG5-GAL4 (DBD)-hCAR/hPXR (LBD)-puromycin plasmid and stable clones were selected in the presence of 0.5 μg/ml puromycin[54]. Cell lines were seeded at a density of 40,000 cells per well in white opaque 96-well plates and treated with test compounds after 24 h. Following incubation with test compounds, cells were incubated for 16 h, the culture medium replaced with test medium containing 0.3 μM luciferin solution and fluorescence quantitated.

### Transient transfection in HepG2 Cells
HepG2 cells (ATCC, HB-8065) seeded in 24-well plates were transfected with CYP2B6-2.2 kb reporter construct[55] containing both a PB-responsive enhancer module and the distal XREM in the presence of hCAR1+A expression vector[30] and pRL-TK renilla luciferase vector (Promega) using X-tremeGENE 9 DNA Transfection Reagent (Roche Diagnostics Corporation, Indianapolis, IN) before treatment with 0.1% DMSO, CITCO (1 μM), DIM or DIE (1, 5, and 10 μM) for another 24 h. Activities of both firefly and renilla luciferases were detected using a Dual-Luciferase Kit (Promega).

### H4IIE stable reporter gene assays
H4IIE-luc bioassays were performed according to previously reported methods[56]. Briefly, trypsinized cells (20,000 cells per well) were seeded into the 60 interior wells of 96 well plates. After 24 h incubation, test and control wells were dosed with test compounds for 72 h and luciferase luminescence quantified. Responses of the H4IIE-luc bioassay (expressed as mean relative luminescence units) were converted to fold changes of the vehicle dosed group.

### Primary human hepatocyte (PHH) assay
PHH culture, RNA extraction and gene expression analyses were performed as described previously[57]. Primer sequences for *cyp2b6*, *cyp3a4*, *sult2a1*, and glyceraldehyde-3-phosphate dehydrogenase (*gapdh*) are shown in Supplementary Table 3. Expression values were quantified using the $2^{-\Delta\Delta Ct}$ method, normalizing to *gapdh* as the housekeeping gene and determining the fold change compared to control values ($n = 3$).

24 h after seeding at $0.25 \times 10^6$ cells/well in 24-well biocoat plate, PHHs were infected with 2 μL of adenoviral enhanced yellow fluorescent protein-tagged hCAR (Ad/EYFP-hCAR) followed by treatment with 0.1% DMSO, phenobarbital (PB) (1 mM), DIM (10 μM) or DIE (10 μM) for 8 h. Cells were washed with phosphate-buffered saline and fixed for 30 min with 4% formaldehyde at room temperature and then stained with 4,6-diamidine-2-phenylindole dihydrochloride for 30 min after washing three times with phosphate-buffered saline. The subcellular localization of hCAR and nuclei were visualized using a Nikon Eclipse Ti-E inverted fluorescence microscope (Nikon, Edgewood, NY)

in YFP/FITC and DAPI fluorescence channels. Compounds were added to the plate in triplicate and results analyzed as detailed below with z-factors calculated as previously described[57].

### Histology
Liver tissues were fixed in 10% neutral-buffered formalin for 24 h, processed and embedded in paraffin. Liver tissue sections were made at 5 μm thickness and deparaffinized in xylene and ethanol. Sections were stained with hematoxylin and eosin (H&E) as per the manufacturer's protocol.

### Statistics and reproducibility
One-way ANOVA tests with multiple comparisons tests (for >3 groups) and two-tailed unpaired *t*-tests (for 2 groups) were performed as specified in figure legends, using Prism 9 (GraphPad Software, Inc., La Jolla, CA). All data were shown as means ± SEM or means ± SD as specified in the figure legends. Data were considered to be significant if $p < 0.05$. At least three independent experiments were performed in pulldowns, TSA, luciferase reporter assays, and Lactobacillus diindole production assays. Two independent experiments were performed in ITC, HDX-MS, TR-FRET, and subcellular localization assays. Primary human hepatocyte CAR subcellular localization assays were randomly allocated into experimental groups. Images were unbiasedly collected and analyses performed by blinded analyses. All attempts at replication were successful. For mouse experiments, sample size was determined based on previous studies[5,51,58,59]. All biological replicates supporting reproducibility are described in each figure legend. No data were excluded in other analyses.

### Reporting summary
Further information on research design is available in the Nature Portfolio Reporting Summary linked to this article.

## Data availability
Metabolomic data generated in this study have been deposited in MassIVE under accession code ID: MSV000091840 (https://doi.org/10.25345/C5NV99M7M). Protein structure used in this study has been deposited in PDB under code 1XVP

## Code availability
The code used in the pulldown analysis is available from https://github.com/huiUofT/humanCAR[60].

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

## Acknowledgements

We thank K.Doty of the Histology laboratory, UIUC for assistance in setting up experiments, and X.Wang for providing human gut microbe extracts. J.L. was supported by a Charles H. Best Postdoctoral Fellowship and Precision Medicine Initiative (PRiME) Fellowship. The work was supported by a Canadian Institutes of Health Research (PJT-186117) to H.M.K., a New Frontiers in Research Fund grant (NRFRE-2019-00901) to H.M.K. and H.P., National Institutes of Health grants (R01CA222469) to S.M., (R01DK113080) to S.A., (R01CA262084) to L.L. and (R21AA02852) to H.W., an American Cancer Society grant (RSGACS132640) to S.A., a Natural Sciences and Engineering Research Council of Canada (NSERC) Discovery grants (RGPIN480432, and CRDPJ597036) to D.J.W. and an Agence Nationale de la Recherche SYNERACT grant to P. B. Infrastructure support was also provided by the Canada Foundation for Innovation, Ontario Research Fund, and NSERC Research Tools and Instrument grants.

## Author contributions

J.L., S.M., P.B., D.W., W.N., H.W., H.P., S.A., and H.K. designed the experiments. J.L. and A.M. performed the protein expression, characterization, pulldown, and ligand identification experiments shown in Fig.1 and Supplementary Figs. 1–3, and performed the experiments in Fig. 4a–f and Supplementary Fig. 3. J.L. also performed isolations, purifications, and characterizations depicted in Fig. 1, Supplementary Fig. 2 and Table 1, and performed the experiments shown in Fig. 2 and Supplementary Fig. 5. A.M. also performed experiments depicted in Fig. 5b and Supplementary Fig. 6a. V.C. performed the HDX experiments in Fig. 3g, h. L.L. performed the experiments in Fig. 4a, c, d. M.G. performed the experiments in Supplementary Fig. 6b, c. H.L. performed the experiments in Supplementary Fig. 6d. H.B. and J.S. performed the experiments in Supplementary Fig. 6e. A.A. and N.D. performed the experiments in Fig. 5 and Supplementary Fig. 7, 8, and 10. J.C. performed the experiments in Supplementary Fig. 9. J.L. and A.M. wrote the initial manuscript. H.M.K. revised and edited the manuscript.

## Competing interests

The authors declare no competing interests.
