## [Peer Review File · Nature Communications]

REVIEWER COMMENTS

Reviewer #1 (Remarks to the Author):

Liu et al. describe a series of diindole compounds produced by commensal bacteria as new agonist ligands for the nuclear receptor CAR. These compounds were elegantly identified by mass spectrometry and 2D NMR using a His-tagged CAR LBD-SRC1 fusion protein to select candidate selective ligands from an extract derived from a complex mixture of microbes cultured from human stool samples. Sophisticated biochemical studies confirm diindole binding to CAR in vitro. Functional studies show that the diindoles act as CAR agonists in cell based assays. Consistent with these results, two of these diindoles, diindolymethane (DIM) and diindolyethane (DIE), upregulate CAR target gene expression in human primary hepatocytes and mouse liver. In contrast to the potent synthetic mouse CAR agonist TCPOBOP, however, the diindoles did not induce liver growth, which is ascribed to “hyperactive” activation of CAR by TCPOBOP. Similarly, the response of human CAR to the diindoles is distinct from that of the synthetic agonist CITCO. While it is clear that TCPOBOP is powerful and stable agonist, the mouse CAR activator phenobarbital is also able to induce a proliferative response that is thought to be due entirely to the constitutive activity of mCAR. Thus, the weaker impact of the diindoles could be due to dosing or other pharmacokinetic issues or simply more selective target gene activation, and the emphasis on overactivation by TCPOBOP seems overdrawn. Future studies will be required to more clearly define the distinct target genes of the diindoles relative to the synthetic agonists.

Reviewer #2 (Remarks to the Author):

This manuscript describes the identification and characterization of a set of indole dimers, referred to as diindoles, as potent CAR agonists. The results are very interesting, however, there are inconsistencies and a lack of important details during the structural characterization of these compounds.

Pull-down experiments coupled to LC-MS revealed target ligands of CAR at $m/z = 130.0652$, 144.0808, 158.0964, 172.1120 and 186.1277. The authors started the structural elucidation with the smallest, $m/z = 130.0$, which was discarded as 3-methylene-indoline or quinoline. While the evidence for quinoline is convincing (different RT), this is not the case with 3-methylene-indoline. This compound is commercially available and should be experimentally rejected using retention time and MS2 data.

After discarding another possibility based on $[M+H-H_2O]^+$, the authors postulated that the enriched candidates from the pull-down experiment result from in-source fragmentation (ISF) events during ESI. Although this is a valid and plausible scenario, the development of downstream arguments and experiments is weak. For example:

(i) Among all the possible compounds with an ISF m/z 130.0652, how DIM was assumed? (details in reference 19 are not even high resolution MS). Could you explain the reasons which lead to DIM?

(ii) Fig 1d shows the MS/MS spectrum of the standard DIM and the enriched ligand from the pull-down. Surprisingly, the parent ion at m/z 247.1229 is the most intense ion in the mass spectrum. I cannot read the collision energy anywhere in the manuscript, however, one would expect that a full scan MS1 acquisition will never produce higher (in-source) fragmentation than a truly MS/MS experiment. Consequently, it is very likely that m/z 247.1229 is detected with the same RT as m/z 130.0652 in the pull-down experiments (Fig 1a and 1b). Have you looked for m/z 247.1229?

(iii) The voltage used in ESI can certainly induce ISF, however, this voltage can be modulated. Considering that standard DIM is available, I miss a full scan MS1 experiment under the same identical ionization conditions (and modifications of the original value) to prove that m/z 130.0652 is the main ion in the spectrum, which doesn't seem to be the case from the MS2 acquisition.

(iv) The authors state that a scaled-up isolation of the metabolites from crude human gut microbiome (GM) samples was performed based on their previously observed retention times and masses. Why the EICs in Fig 1c do not match with those in Fig 1e? Have you used different chromatographic (gradient) conditions? If so, please use the same conditions for consistency. It doesn't make sense that all five candidates are separated within a small range of less than one minute in Fig 1c, while in Fig 1e this range is 10 minutes approximately.

The scaled-up isolation should be used to explore and validate that ionization behaviour of all candidates, both at the MS1 (in-source fragments?) and MS2 level. This is missing.

(v) The authors state that the comparison of the retention times of the affinity-purified and custom purified molecules with those identified by affinity pull-downs and mass spectrometry confirmed the identities of all suspected diindoles. However, the MS1 and MS2 data of these comparisons are missing.

(vi) Extended Data Fig. 2a: how do the authors explain that I3C and quinoline present an EIC m/z 130 at the same retention time (6 minutes approximately) as DIM. Clearly it is a smaller ion, but still coincides with it.

Other points:

- The authors demonstrate that the diindoles are generated in a pH-dependent manner from indole and corresponding aldehydes via sequential Friedel-Crafts reactions. This is a very interesting observation. The quality of the paper would increase significantly if the levels of indole and aldehydes are studied from crude human gut microbiome, and whether these can be manipulated to alter the production of diindoles.

- Line 272: it is unlikely that DIE is produced in the liver at low pH. Under which conditions the pH of the liver can be so low?

- Line 487: Details of the LC-MS and LC-MS/MS analysis are missing, and it is not acceptable to cite ref 51 because the conditions and experiments are not the same.

- Title & Abstract: the descriptions 'microbiota-derived diindoles' and 'diindole molecules produced by commensal bacteria' are insufficiently rigorous because diindoles are not produced by the microbiota per se through enzymatic reactions. If only the authors had investigated the production of indole and aldehydes by the microbiota and how they correlate with diindole levels, these descriptions would not sound so inaccurate.

Minor point:

- Abstract: consider using the abbreviations DIM and DIE

Reviewer #3 (Remarks to the Author):

Liu J and colleagues submitted an original manuscript reporting the discovery of different diindoles molecules derived by intestinal commensal bacteria that function as agonists for the CAR. The work has been done mainly in vitro and the data are very clearly and rigorously presented and the experiments run with proper controls. I particularly appreciate the experiments in Figure 5 and extended Figure 6 where the authors gave the diindoles in vivo in mouse models, WT or Car KO and convincingly showed the upregulation of Car in the liver of only WT animals.

The identification of diindoles from ex vivo intestinal samples was conducted via mass spec on the crude extract of the human gut microbiota samples. I was wondering here if the authors controlled for dietary presence of the diindoles. Probably this would be easily to be conducted in animals: could the authors measure the diindoles content in normal mouse diets and in the murine intestinal content of mice colonised with a microbe vs germ-free mice fed the same diet as the colonised mice? In this way they will be sure that the diindoles present in the intestinal content come only from the bacteria.

Perhaps these data are already available from public databases.

Could also the authors perhaps interrogate Human Microbiome Database and search among the many available metagenomics and metatranscriptomics data published which type of bacteria would be responsible for the diindoles production?

I would be interested in seeing how these compounds could be used in vivo in different preclinical models of diseases, but I completely understand that this will be the task for a follow-up project.

CAR Manuscript_Response to reviews

Reviewer #1

Liu et al. describe a series of diindole compounds produced by commensal bacteria as new agonist ligands for the nuclear receptor CAR. These compounds were elegantly identified by mass spectrometry and 2D NMR using a His-tagged CAR LBD-SRC1 fusion protein to select candidate selective ligands from an extract derived from a complex mixture of microbes cultured from human stool samples. Sophisticated biochemical studies confirm diindole binding to CAR in vitro. Functional studies show that the diindoles act as CAR agonists in cell based assays. Consistent with these results, two of these diindoles, diindolylmethane (DIM) and diindolyethane (DIE), upregulate CAR target gene expression in human primary hepatocytes and mouse liver.

In contrast to the potent synthetic mouse CAR agonist TCPOBOP, however, the diindoles did not induce liver growth, which is ascribed to “hyperactive” activation of CAR by TCPOBOP. Similarly, the response of human CAR to the diindoles is distinct from that of the synthetic agonist CITCO. While it is clear that TCPOBOP is powerful and stable agonist, the mouse CAR activator phenobarbital is also able to induce a proliferative response that is thought to be due entirely to the constitutive activity of mCAR. Thus, the weaker impact of the diindoles could be due to dosing or other pharmacokinetic issues or simply more selective target gene activation, and the emphasis on overactivation by TCPOBOP seems overdrawn. Future studies will be required to more clearly define the distinct target genes of the diindoles relative to the synthetic agonists.

We thank the reviewer for their positive review, comments and suggestions. Regarding the TCPOBOP response to diindoles, we agree that we may have oversimplified somewhat. The difference we saw in LBD structure, physical interaction data and transcriptional assay responsiveness all suggested differences in binding and outcomes that were most easily read as lower levels of activation, though differences in cofactor type binding or levels could also lead to differences in target gene activation levels or selectivity. Consequently, we have changed our wording to more completely encompass these varying mechanisms and outcomes.

Reviewer #2

This manuscript describes the identification and characterization of a set of indole dimers, referred to as diindoles, as potent CAR agonists. The results are very interesting, however, there are inconsistencies and a lack of important details during the structural characterization of these compounds.

Pull-down experiments coupled to LC-MS revealed target ligands of CAD at $m/z = 130.0652$, 144.0808 , 158.0964 , 172.1120 and 186.1277 . The authors started the structural elucidation with the smallest, $m/z = 130.0$, which was discarded as 3-methylene-indoline or quinoline. While the evidence for quinoline is convincing (different RT), this is not the case with 3-methylene-indoline. This compound is commercially available and should be experimentally rejected using retention time and MS2 data.

We opted not to pursue 3-methylene-indoline for several reasons: 1) it exhibits relatively high polarity ($\log P = -3.58$ calculated by EPI Suite) compared to diindoles (*e.g.*, $\log P = 4.37$ for DIM), making it unlikely to elute with 80~90% MeOH; and 2) despite being commercially available, it is extremely expensive; 3) We were highly encouraged by the observation of in-source fragments and corresponding molecular ions at larger mass ranges; 4) In the end, it was not a factor as purchased or synthesized diindole controls proved to have all of the same properties as the affinity purified compounds.

After discarding another possibility based on $[M+H-H_2O]^+$, the authors postulated that the enriched candidates from the pull-down experiment result from in-source fragmentation (ISF) events during ESI. Although this is a valid and plausible scenario, the development of downstream arguments and experiments is weak. For example:

(i) Among all the possible compounds with an ISF m/z 130.0652, how DIM was assumed? (details in reference 19 are not even high resolution MS). Could you explain the reasons which lead to DIM?

ESI is typically considered as ‘soft ionization’, thus we hadn’t expected the observation of such abundant in-source fragments. We invested significant time and effort to assign the structures of these tentative ligands. Initially, elemental analysis of the MS^1 mass feature revealed that all of the molecules are nitrogen-containing compounds, which might be indole compounds based on the characteristic MS^2 fragment at m/z 130.0652. Subsequently, we conducted a thorough search for structures with these characteristics using SciFinder, focusing on the indole derivatives in natural product occurrence and in food or animal feed. After ruling out possibilities such as quinoline, 3-methylene-indoline and indole-3-carbinol, we considered that the detected ions might be derived from a neutral loss. We didn’t rely on the data reported in the Ref. 19, but the findings did support the hypothesis that a neutral indole of DIM may be lost due to ISF. These steps led us to purchase a DIM standard, which compared identically in ionization behavior and retention time to the extracted compound.

(ii) Fig 1d shows the MS/MS spectrum of the standard DIM and the enriched ligand from the pull-down. Surprisingly, the parent ion at m/z 247.1229 is the most intense ion in the mass spectrum. I cannot read the collision energy anywhere in the manuscript, however, one would expect that a full scan MS^1 acquisition will never produce higher (in-source) fragmentation than a truly MS/MS experiment. Consequently, it is very likely that m/z 247.1229 is detected with the same RT as m/z 130.0652 in the pull-down experiments (Fig 1a and 1b). Have you looked for m/z 247.1229?

We thank the reviewer for pointing out the unclear annotation in Fig. 1d. The unlabeled mass in the previous figure was not the m/z 247.1229 molecular ion but rather a co-eluting (245.1067) ion. A blowup of this region is attached in ‘**Response Figure 1**’ attached below. The co-eluting m/z 245.1067 was further excluded by checking its chromatographic profile with the MS^2 fragment 130.0652. Note that the m/z 247.1223 peak is much weaker than the m/z 130.0649 peak. We also show in **Response Figure 2** that the 245.1067 mass does not go away in the MS^2 spectra, whereas the m/z 247.1223 peak is further diminished. We have clarified this point in the revised figure and legend.

(iii) The voltage used in ESI can certainly induce ISF, however, this voltage can be modulated. Considering that standard DIM is available, I miss a full scan MS1 experiment under the same identical ionization conditions (and modifications of the original value) to prove that m/z 130.0652 is the main ion in the spectrum, which doesn't seem to be the case from the MS2 acquisition.

This is shown in **Response Fig. 1** below. We have now included this as Supplemental figure 2c.

(iv) The authors state that a scaled-up isolation of the metabolites from crude human gut microbiome (GM) samples was performed based on their previously observed retention times and masses. Why the EICs in Fig 1c do not match with those in Fig 1e? Have you used different chromatographic (gradient) conditions? If so, please use the same conditions for consistency. It doesn't make sense that all five candidates are separated within a small range of less than one minute in Fig 1c, while in Fig 1e this range is 10 minutes approximately. The scaled-up isolation should be used to explore and validate that ionization behaviour of all candidates, both at the MS1 (in-source fragments?) and MS2 level. This is missing.

The differences noted by the reviewer were due to differences in the initial analysis and semi-preparative column sizes and compositions. As detailed in the experimental methods section, the chromatogram in Fig 1e was generated using a $5\ \mu\text{m}$ $100\ \text{\AA}$, $250 \times 10\ \text{mm}$ C8 column, while the chromatogram in Fig 1c by a C18+, $1.5\ \mu\text{m}$ $50 \times 2.1\ \text{mm}$ column. The semi-preparative column's efficiency and resolution are notably lower than those of the UPLC analysis column such that the UPLC analysis column facilitated the separation of the five candidates over a small range, while the semi-preparative column required the much longer 10-minute separation.

The isolated compounds underwent additional validation at MS1, and the spectra are presented below in Response Fig. 3. While we did not explore their ionization behaviors, the method was sufficient for detection and quantification. Note that the compounds purified from the semi-preparative columns were analyzed using the same UPLC column and conditions, and thus the retention time and MS1 spectra were comparable. We have now included the EICs as Supplemental figure 3.

(v) The authors state that the comparison of the retention times of the affinity-purified and custom purified molecules with those identified by affinity pull-downs and mass spectrometry confirmed the identities of all suspected diindoles. However, the MS1 and MS2 data of these comparisons are missing.

We have added an Extended figure to the manuscript that shows these comparisons of MS1 data as the MS2 data were not applicable due to the reason noted above.

(vi) Extended Data Fig. 2a: how do the authors explain that I3C and quinoline present an EIC m/z 130 at the same retention time (6 minutes approximately) as DIM. Clearly it is a smaller ion, but still coincides with it.

The peak was not clearly labeled in the old version. In the updated version, the major peak is labeled as I3C. The minor peak might be DIM converted from the I3C pathway (shown in **Response Fig. 4**).

Other points:

- The authors demonstrate that the diindoles are generated in a pH-dependent manner from indole and corresponding aldehydes via sequential Friedel-Crafts reactions. This is a very interesting observation. The quality of the paper would increase significantly if the levels of indole and aldehydes are studied from crude human gut microbiome, and whether these can be manipulated to alter the production of diindoles.

We thank the reviewer for appreciating the novelty and significance of this finding. In the gut, this reaction depends on both the production of indoles and aldehydes by the bacteria (or in diet) as well as low pH. Replicating these conditions with a crude microbiome culture can be tricky. That said, we were able to show that monocultures of lactobacilli supplemented with indole are able to generate diindoles (Supp. Fig. 9). In addition, in animal studies, we have demonstrated that diindoles can be detected in mouse intestinal content, although their levels exhibit variability among mice (**Response Fig. 5a**). This variability is likely attributed to differences in diet, cage bedding and gut bacteria compositions (PMID: 33257713). We also show here that they accumulate in the mouse liver and elsewhere in the intestine when provided orally and when endogenous CAR is knocked out (**Response Fig. 5b and 5c, new Supp. Fig. 10**).

- Line 272: it is unlikely that DIE is produced in the liver at low pH. Under which conditions the pH of the liver can be so low?

We were not implying that the pH of liver is low. Our results showed that significant levels of DIE can be produced at pH 7.4 (physiological pH) after long incubations of acetaldehyde and indole. This suggested the possibility that diindoles could be produced under abnormal conditions where acetaldehyde and indole levels are high.

- Line 487: Details of the LC-MS and LC-MS/MS analysis are missing, and it is not acceptable to cite ref 51 because the conditions and experiments are not the same.

Thank you for pointing this out. The relevant details have been added to the revised manuscript.

- Title & Abstract: the descriptions ‘microbiota-derived diindoles’ and ‘diindole molecules produced by commensal bacteria’ are insufficiently rigorous because diindoles are not produced by the microbiota per se through enzymatic reactions. If only the authors had investigated the production of indole and aldehydes by the microbiota and how they correlate with diindole levels, these descriptions would not sound so inaccurate.

Thanks for pointing out the nuances and potential misinterpretation. We have changed the title and abstract accordingly. For example, the new title is “Diindoles produced from commensal microbiota metabolites function as endogenous CAR ligands”.

Minor point:

- Abstract: consider using the abbreviations DIM and DIE

We changed the abstract to include diindolylmethane (DIM) and diindolyethane (DIE).

Reviewer #3

Liu J and colleagues submitted an original manuscript reporting the discovery of different diindoles molecules derived by intestinal commensal bacteria that function as agonists for the CAR. The works has been done mainly in vitro and the data are very clearly and rigorously presented and the experiments run with proper controls. I particularly appreciate the experiments in Figure 5 and extended Figure 6 were the authors gave the diindoles in vivo in mouse models, WT or Car KO and convincely showed the upregulation of Car in the liver of only WT animals.

The identification of diindoles from ex vivo intestinal samples was conducted via mass spec on the crude extract of the human gut microbiota samples. I was wondering here if the authors controlled for dietary presence of the diindoles. Probably this would be easily to be conducted in animals: could the authors measure the diindoles content in normal mouse diets and in in the murine intestinal content of mice colonised with a microbes vs germ-free mice fed the same diet as the colonised mice? In this way they will be sure that the diindoles present in the intestinal content come only from the bacteria.

From our investigations thus far, diindoles are not generally present in our major dietary sources and mouse diet. However, their precursors are widely present. Their presence is largely dependent on bacteria (PMID: 35505660). We don't yet know of any bacteria that produce significant levels of diindole on their own, but there are many in the literature that have already been shown to produce high and effective levels of indole (PMID:27916477) as well as adjust pH in small intestine (PMID: 35505660). There are also many bacteria that can produce many types of aldehydes from different endogenous or dietary precursors (PMID: 12755455). Aldehydes can also be abundant in diet (PMID: 31368096). All of this is documented in the manuscript. As we also show in the manuscript, any time the combination of aldehyde and indole coincide in the low pH environment in the upper intestine, they are efficiently and spontaneously converted into diindoles.

While diindoles can be detected in mouse intestinal content, their levels exhibit variability among mice (**Response Fig. 5a**). This variability is likely attributed to differences in diet, cage bedding, and gut bacteria compositions (PMID: 33257713). We also show here that they accumulate in the mouse liver and elsewhere in the intestine when provided orally and when endogenous CAR is knocked out (**Response Fig. 5b and 5c, new Supp. Fig. 10**). Thus, the diindoles are reaching the liver and activating CAR, which then activates a negative feedback loop. This activity tends to keep circulating levels below detection via LC-MS. It has been shown that DIM can be metabolized by P450, SULT, and UGT (PMID: 34035125). While it is likely that these levels spike in the intestine and liver after appropriate meals, it would take considerably more time to explore the many variables involved.

Perhaps these data are already available from public databases.

As we showed in this study, the parent ions of diindoles are highly unstable in the ESI source under most conditions, with rapid conversion into masses identical to other molecules, such as I3C, IPA, etc. The latter are present in public databases but are impossible to equate with diindole levels.

Could also the authors perhaps interrogate Human Microbiome Database and search among the many available metagenomics and metatranscriptomics data published which type of bacteria would be responsible for the diindoles production?

See also our previous answer to the bacterial sources for diindoles. We show that they can contribute indirectly via the production of indole and aldehydes. As we shown, lactobacilli cultured with indole-containing media produce diindoles, both by providing aldehydes and by lowering the pH of their environment. Previous publications have identified many gut bacteria that produce indoles converted from tryptophan (PMID:27916477). As a supplementary observation, we have demonstrated that *Clostridium sporogenes*, a major indole producer, can also produce DIM and DIE when cultured with tryptophan-containing media, with exhibiting higher production levels (see Response Fig. 6) compared with *Lactobacillus plantarum*. Consequently, it is highly plausible that the indole-producing, aldehyde-producing, and pH-adjusting bacteria collectively contribute to diindole production.

I would be interested in seeing how these compounds could be used in vivo in different preclinical models of diseases, but I completely understand that this will be the task for a follow-up project.

We totally agree and are working on such studies. These prospects have been included in the discussion section of the revised manuscript.

Response Figure 1. **a,b** show the MS1 spectra range from m/z 90–250 for the CAR-extracted ligand (above) and pure DIM (purchased, below). **c,d** show zoomed in views of the red boxed region shown in **a** and **b**.

Response Fig. 2. a,b show MS2 spectra in the range from m/z 50–250. c,d zoomed in view of the red box shown in a and b.

Response Fig. 3. MS1 spectra of diindole standards. The top panel shows the experimental spectrum and the bottom a simulation.

Response Fig. 4. I3C pathway for generating DIM.

Response Figure 5. The levels of diindoles in male mouse tissues. a) Endogenous diindoles are present in the mouse gastrointestinal tissues. **b, c)** DIM and DIE were increased in mouse liver and intestine when endogenous CAR is knockout.

Response Figure 6. *Lactobacillus plantarum* (LP) and *Clostridium sporogenes* (CS) generate DIM and DIE with tryptophan supplementation. There are no DIM and DIE are present in the media. LP and CS and utilize the tryptophan in the media to produce DIM and DIE.

REVIEWER COMMENTS

Reviewer #2 (Remarks to the Author):

The authors have partially addressed the main points of the revision.

However, there's still a couple of points that need to be clarified:

- Fig 1d: How is it possible that m/z 245.1067 is present in the standard DIM spectrum? What kind of interference/contamination is this?

- Supplemental figure 2c is not addressing the point raised in the previous revision: I'm asking to use the pure standard to unequivocally demonstrate the ISF events at different ESI voltages. This should be a very easy task. This is particularly relevant because DIM has been extensively analyzed by LC-ESI MS/MS in plasma and urine samples, and it doesn't seem to suffer such level of in-source fragmentation.

Reviewer #3 (Remarks to the Author):

Thanks to the authors for providing a comprehensive revision of their manuscript, they answered to all my comments in a satisfactory manner.

Once again we appreciate the diligence of the second reviewer, who remains somewhat concerned about the relative lack of diindole parent ion masses in our LC-MS analyses. They also continued to wonder about the nature of the m/z 245.1067 peak. While we were not concerned ourselves about these, given the many other validations and explanations, we were happy to take advantage of the opportunity to delve further into the issue. Our findings and answers, highlighted in blue below, will likely prove helpful for other researchers interested in diindole metabolomics.

Reviewer #2 (Remarks to the Author):

The authors have partially addressed the main points of the revision.

However, there's still a couple of points that need to be clarified:

- Fig 1d: How is it possible that m/z 245.1067 is present in the standard DIM spectrum? What kind of interference/contamination is this?

Our analyses below confirm that m/z 245.1067 is a fragment of the DIM parent ion resulting from the expedient neutral loss of H_2 . This results in the formation of a highly stable product with sequential conjugate double bonds (diagramed in response Fig. 1 below). To confirm this, we synthesized and tested a deuterium-labeled version of DIM (response Fig. 2a). The observed fragmentation pattern closely matched that of the DIM standard but with the expected parent ion shift to 249.1335 ($C_{17}H_{13}D_2N_2$) and breakdown products resulting from the loss of either D or H (m/z : 248.1277, 247.1191 and 246.1137 shown in response Fig. 2a). The products support the loss of D from the middle carbon and the formation of a double bond. To further explore the fragmentation mechanism, we collected MS^2 spectrum of each in source fragment including m/z 249.1356, 248.1277, and 247.1191 (response Fig. 2b). Unexpectedly, the fragment of m/z 246.1134 was only detected from the MS^2 spectrum of the radical ion 248.1277. Accordingly, we propose the fragmentation mechanism shown in response Fig. 2c.

Supplemental figure 2c is not addressing the point raised in the previous revision: I'm asking to use the pure standard to unequivocally demonstrate the ISF events at different ESI voltages. This should be a very easy task. This is particularly relevant because DIM has been extensively analyzed by LC-ESI MS/MS in plasma and urine samples, and it doesn't seem to suffer such level of in-source fragmentation.

We have done as the reviewer asked. First, we checked the literature for evidence of ESI-MS/MS data on DIM that shows lower levels of fragmentation, but did not find any (see for example PMID: 34875720, 34035125). We also found a similar fragmentation tendency for another indole-containing compound, tryptophan, which yields a radical cation through collision-induced dissociation (CID) involving a redox-active metal ternary complex (Phys. Chem. B 2004, 108, 30, 11170–11181). Another investigation, using copper-mediated low-energy CID conditions, also observed a predominant product—a 3-methyleneindolum ion at m/z 130 (PMID: 23512424).

We then conducted a series of experiments subjecting DIM to different spraying voltages. Surprisingly, despite ESI being considered a soft ionization method, we observed ISF for all diindoles even at the lowest spray voltage (response Fig. 3). The unique structure of DIM and the high stability of fragment ions, likely explains the sensitivity to ions and relative product ratios.

We also explored into the stability of the initial molecular ions using different energies in HCD. As with ISF, extensive fragmentation was observed even at the lowest collision energy (10 eV: response Fig. 4).

In summary, these findings further support the notion that, not just DIM, but all of the diindoles identified in this study, are susceptible to ISF, primarily due to the generation of radical ions from

active nitrogen lone pair electrons and the formation of highly stable fragment ions due to this conjugated system.

Response Fig. 1. Proposed fragmentation of DIM to generate the 245.1069 feature.

Response Fig. 2. Validation of DIM fragment ion products using deuterium modified DIM. a) MS1 spectra of DIM-d₂. The top panel exhibit the full spectra ranging from *m/z* 120 to 250, while the bottom panel shows a close-up view of the range *m/z* 245-250.5 in the top panel. b) MS2 spectra of *m/z*: 248.1277, 247.1191 and 246.1137 shown in a). c) Diagram of proposed fragmentation.

DIM

Spray voltage: 3.5 Kv

Spray voltage: 2.5 Kv

Spray voltage: 2 Kv

DIE

Spray voltage: 3.5 Kv

Spray voltage: 2.5 Kv

Spray voltage: 2 Kv

DIP

Spray voltage: 3.5 Kv

Spray voltage: 2.5 Kv

Spray voltage: 2 Kv

DIB

Spray voltage: 3.5 Kv

Spray voltage: 2.5 Kv

Spray voltage: 2 Kv

Response Fig. 3. Comparison of diindole ISF products under different spray voltages.

Diindole_CE_10v #1598 RT: 5.71 AV: 1 NL: 1.07E6
F: FTMS + c ESI Full ms2 247.1229@hcd10.00 [50.0000-525.0000]

Response Fig. 4. Fragmentation of m/z 247.1229 at 10 eV.

REVIEWERS' COMMENTS

Reviewer #2 (Remarks to the Author):

Thanks for the additional details. All is clear now.